# Fe_3_O_4_ Nanozymes Improve Neuroblast Differentiation and Blood-Brain Barrier Integrity of the Hippocampal Dentate Gyrus in D-Galactose-Induced Aged Mice

**DOI:** 10.3390/ijms23126463

**Published:** 2022-06-09

**Authors:** Zihao Xia, Manman Gao, Peng Sheng, Mengmeng Shen, Lin Zhao, Lizeng Gao, Bingchun Yan

**Affiliations:** 1Jiangsu Key Laboratory of Integrated Traditional Chinese and Western Medicine for Prevention and Treatment of Senile Diseases, Medical College, Yangzhou University, Yangzhou 225001, China; chinaxiky@yeah.net (Z.X.); 18752784220@163.com (M.G.); dh15751683333@163.com (P.S.); mmshen666@163.com (M.S.); 15006779450@163.com (L.Z.); 2CAS Engineering Laboratory for Nanozyme, Institute of Biophysics, Chinese Academy of Sciences, Beijing 100101, China; gaolizeng@ibp.ac.cn

**Keywords:** Fe_3_O_4_ nanozyme, neuroblast differentiation, blood–brain barrier, antioxidant, aging

## Abstract

Aging is a process associated with blood–brain barrier (BBB) damage and the reduction in neurogenesis, and is the greatest known risk factor for neurodegenerative disorders. However, the effects of Fe_3_O_4_ nanozymes on neurogenesis have rarely been studied. This study examined the effects of Fe_3_O_4_ nanozymes on neuronal differentiation in the dentate gyrus (DG) and BBB integrity of D-galactose-induced aged mice. Long-term treatment with Fe_3_O_4_ nanozymes (10 μg/mL diluted in ddH_2_O daily) markedly increased the doublecortin (DCX) immunoreactivity and decreased BBB injury induced by D-galactose treatment. In addition, the decreases in the levels of antioxidant proteins including superoxide dismutase (SOD) and catalase as well as autophagy-related proteins such as Becin-1, LC3II/I, and Atg7 induced by D-galactose treatment were significantly ameliorated by Fe_3_O_4_ nanozymes in the DG of the mouse hippocampus. Furthermore, Fe_3_O_4_ nanozyme treatment showed an inhibitory effect against apoptosis in the hippocampus. In conclusion, Fe_3_O_4_ nanozymes can relieve neuroblast damage and promote neuroblast differentiation in the hippocampal DG by regulating oxidative stress, apoptosis, and autophagy.

## 1. Introduction

Brain aging is a complex biological process that is associated with progressive neurological dysfunction and is a significant risk factor for most common neurodegenerative diseases. The classic free-radicals theory of aging postulates that aging is a consequence of the attack on cells and tissues by free radicals. Reactive oxygen species (ROS) including superoxide (O_2_^−^), hydroxyl radicals (·OH), and hydrogen peroxide (H_2_O_2_) are considered to be the most important oxidant molecules in cells [1]. ROS have been implicated in a multitude of biological processes under physiological conditions including inflammation responses, apoptosis, autophagy, and synaptic plasticity as well as learning and memory ability [2]. However, when overproduced, these molecules can induce oxidative stress, which can trigger cell death and is responsible for the pathogenesis of many neurodegenerative diseases [3]. The hippocampus is a major region of the brain that shows progressive age-related increases in protein nitration and oxidation, together with decreases in SOD, catalase, and glutathione (GSH) reductase activity [4]. Neurogenesis in the subgranular zone (SGZ) of the hippocampal dentate gyrus (DG) has been suggested to contribute to brain cognitive reserve and brain plasticity [5,6]. In the aging brain, the suppression of neurogenesis induced by increased oxidative stress promotes the aging process of the nervous system [7,8]. Neural stem cells are considered as key determinants of neurogenesis, and the control of neural stem cell differentiation is a promising approach to manipulate neural cells for therapeutic purposes [9].

Apoptosis and autophagy are two evolutionarily conserved processes that play important roles in cell death and cellular homeostasis under normal physiological conditions [10]. ROS act as signaling molecules to induce autophagosome formation and autophagic degradation. Autophagy, in contrast, serves to reduce oxidative damage and ROS levels through the removal of protein aggregates and damaged organelles [11]. Apoptosis is widely considered the principal mechanism of cell death in mammals. Oxidative stress directly induces apoptosis and causes tissue dysfunction [12]. The loss of autophagic function and the induction of apoptosis have been widely reported in the aging brain [13,14,15,16]. In addition, these processes take place in highly specialized neurogenic niches, requiring an appropriate microenvironment maintained by the blood–brain barrier (BBB) to limit entry of blood-derived products, pathogens, and cells into the brain [17,18]. Breakdown of the BBB plays an important role in cognitive impairment as an early event in the aging human brain that begins in the hippocampus [19]. In aging, oxidative stress exacerbates BBB disruption, and chronic BBB breakdown leads to the accumulation of blood-derived neurotoxic proteins in the central nervous system (CNS), causing progressive neurodegeneration with the loss of neurons mediated by either direct neuronal toxicity or oxidative stress [19,20].

D-galactose (D-gal) animal models have been successfully used to study aging processes or for the screening of anti-aging drugs because the administration of D-gal has been shown to result in several hallmarks of aging including increased oxidative stress, decreased activity of antioxidant enzymes, and mitochondrial dysfunction [21]. Previous studies have shown that D-gal treatment results in the exacerbation of apoptosis and dysfunctional autophagy in the brain [22,23,24].

Over the last decade, Fe_3_O_4_ nanozymes have been found to have attractive properties for various biotechnological applications including biosensing [25], magnetic resonance imaging (MRI) [26], in hyperthermia treatment [27], and in targeted drug delivery [28,29,30]. Fe_3_O_4_ nanozymes possess peroxidase-like activity, converting H_2_O_2_ to ·OH and thus increasing the ROS levels at acidic pH [31], and decomposing H_2_O_2_ into H_2_O and O_2_ under the conditions of neutral and basic pH [32]. They retain their catalase-like activity and have protective effects against increased oxidative stress in cultured cells in vitro and in Drosophila [33]. However, there have been few studies regarding the effects of Fe_3_O_4_ nanozymes on neurogenesis in the aging hippocampus. Therefore, the present study investigated the effects of long-term treatment with Fe_3_O_4_ nanozymes on neuronal differentiation and BBB integrity in the hippocampus of D-gal induced aging mice. We also evaluated the levels of antioxidants, apoptosis, and autophagy after Fe_3_O_4_ nanozyme treatment in the hippocampus of D-gal induced aging mice to explore the protective mechanism in hippocampal neurogenesis.

## 2. Results

### 2.1. Concentration of Fe_3_O_4_ Nanozymes in Mice Brain

PEG-Fe_3_O_4_ nanozymes were observed by scanning electron microscope and transmission electron microscope (Figure 1A). With the prolongation of Fe_3_O_4_ nanozyme intake in the drinking water, the content of iron in the brains of mice showed an increasing trend. The basal iron concentration was 21.27 mg/kg without treatment. After 24 h, 48 h, and 72 h of nanozyme intake, the iron contents were 34.24 mg/kg, 62.85 mg/kg, and 80.74 mg/kg (Figure 1B).

### 2.2. Neuroblast Differentiation in the SGZ

DCX, a microtubule-associated protein present in migrating neuroblasts, is one of the most widely used markers of early neurons [34]. The number of DCX^+^ cells was shown to correlate with the rate of neuroblast differentiation [35,36]. DCX^+^ cells were mainly localized in the SGZ of the hippocampal DG on immunohistochemical staining for this immature neuron/neuronal precursor marker. In the control group, many DCX-immunoreactive cells were well observed in the SGZ (Figure 2A). The number of DCX^+^ cells in the D-gal group was much smaller than that in the control group (Figure 2B). However, the Mel+D-gal and pFe_3_O_4_+D-gal groups showed a significantly increased number of DCX^+^ cells compared to the D-gal group (Figure 2D,E). In contrast, the number of DCX^+^ cells was markedly decreased in the pFe_3_O_4_ group compared to the control group (Figure 2C).

We classified the DCX-immunoreactive dendrites into three types according to the method described previously [37]. In the control group, most of the DCX-immunoreactive cells had dendrites of type “α”, which were thickened, became very long, and projected into the molecular layer (ML) of the DG (Figure 2a). In the D-gal and pFe_3_O_4_ groups, many DCX-immunoreactive cells had dendrites of types “β” and “γ” (Figure 2b,c). However, most of the DCX-immunoreactive cells had dendrites of types “α” and “β” in the Mel+D-gal and pFe_3_O_4_+D-gal groups (Figure 2d,e).

### 2.3. Cell Proliferation in the SGZ

Ki-67 is a nuclear protein that is expressed in all phases of the cell cycle except quiescence and is used as a proliferation marker during the initial stages of adult neurogenesis [38]. In the control group, many Ki-67-immunoreactive cells were well observed in the hippocampal DG (Figure 3A). The number of Ki-67^+^ cells in the D-gal group was much smaller than that in the control group (Figure 3B). However, the Mel+D-gal and pFe_3_O_4_+D-gal groups showed a significant increase in the number of Ki-67^+^ cells compared to the D-gal group (Figure 3C,E). In contrast, the number of Ki-67^+^ cells was markedly decreased in the pFe_3_O_4_ group compared to the control group (Figure 3D).

### 2.4. Changes in BBB Integrity

To determine the effects of PEG-Fe_3_O_4_ nanozymes on BBB integrity in the hippocampus, we examined the changes in the PECAM-1 expression by immunohistochemistry and tight junction (TJ)-associated protein levels by Western blotting.

#### 2.4.1. Expression of PECAM-1 in the Hippocampal DG Region

PECAM-1 immunoreactivity cells was readily detected in the hippocampal DG region in the control group (Figure 4A). In the D-gal group, there was a significant decrease in the PECAM-1 immunoreactivity cells in the hippocampal DG region compared to the control group (Figure 4B). However, in the Mel+D-gal and pFe_3_O_4_+D-gal groups, the PECAM-1 immunoreactivity cells was markedly increased compared to the D-gal group (Figure 4C,E). In the pFe_3_O_4_ group, PECAM-1 immunoreactivity cells were decreased compared to the control group (Figure 4D).

#### 2.4.2. Claudin5 and ZO-1 Protein Levels in the Hippocampus

Western blotting analyses showed that the levels of Claudin5 and ZO-1 protein expression were markedly decreased in the D-gal group compared to the control group. However, their levels were markedly increased in the Mel+D-gal and pFe_3_O_4_+D-gal groups compared to the D-gal group. Compared to the control group, both the Claudin5 and ZO-1 protein levels were significantly decreased in the pFe_3_O_4_ group (Figure 5).

### 2.5. Long-Term Treatment with PEG-Fe_3_O_4_ Nanozymes Induces Changes in Oxidants and Antioxidants in the Hippocampus

Malondialdehyde (MDA) content and superoxide dismutase (SOD) activity as indicators of free radical damage maintain the balance between oxidants and antioxidants in normal brain tissue [36,39]. Our results showed that the content of MDA was obviously higher in the D-gal group than in the control group, however, the content of MDA was lower in the Mel+D-gal and pFe_3_O_4_+D-gal group than in the D-gal group (Figure 6A). In addition, we found that the activity of SOD was lower in the D-gal group than in the control group, and the activity of SOD in the Mel+D-gal and pFe_3_O_4_+D-gal group was higher than that in the D-gal group (Figure 6B). Western blotting analysis showed differences in the levels of SOD1, SOD2, and catalase in the hippocampal DG between the groups (Figure 6C). The protein levels of these antioxidants were markedly decreased in the D-gal group compared to the control group. However, their levels were markedly increased in the Mel+D-gal and pFe_3_O_4_+D-gal group compared to the D-gal group. Compared to the control group, the SOD1, SOD2, and catalase protein levels were significantly decreased in the pFe_3_O_4_ group (Figure 6C–F).

### 2.6. Long-Term Treatment with PEG-Fe_3_O_4_ Nanozymes Induces Changes in Cleaved Caspase-3 and Bcl-2 Protein Levels in the Hippocampus

The results showed that long-term D-galactose treatment significantly induced apoptosis in the hippocampus by obviously increasing the cleaved caspase-3 and caspase-3 expression as well as decreasing the Bcl-2 expression compared to the control group (Figure 7). In the Mel+D-gal and pFe_3_O_4_+D-gal groups, the apoptosis induced by D-galactose was inhibited by reducing the levels of cleaved caspase-3 protein expression and increasing the levels of Bcl-2 protein expression. However, the pFe_3_O_4_ group showed markedly increased caspase-3 protein levels and markedly reduced Bcl-2 protein levels compared to the control group (Figure 7).

### 2.7. Long-Term Treatment with PEG-Fe_3_O_4_ Nanozymes Induces Changes in the Protein Levels of Beclin-1, LC3II/I and Atg7 in the Hippocampus

To determine the effects of PEG-Fe_3_O_4_ nanozymes on autophagy in the hippocampus, we examined the changes in the Beclin-1, LC3II/I and Atg7 levels (Figure 8). The levels of Beclin-1, LC3II/I, and Atg7 expression were markedly decreased in the D-gal group compared to the control group, with almost no expression of Atg7 detected in the D-gal group. In the Mel+D-gal and pFe_3_O_4_+D-gal groups, the levels of Beclin-1, LC3II/I, and Atg7 protein expression were significantly increased compared to the D-gal group. However, the pFe_3_O_4_ group showed marked reductions in the levels of these proteins compared to the control group (Figure 8).

### 2.8. Effects of Long-Term Treatment with PEG-Fe_3_O_4_ Nanozymes on the Akt/mTOR Signaling Pathway

The Akt/mTOR signaling pathway plays a pivotal role in stem cell aging and has important effects on the autophagy levels. We found differences in the p-Akt/Akt and p-mTOR/mTOR ratios between the control and experimental groups (Figure 9). Compared to the control group, the ratios of p-Akt/Akt and p-mTOR/mTOR were significantly increased in the D-gal group to restrain autophagy. Furthermore, the Mel+D-gal and pFe_3_O_4_+D-gal groups showed reduced p-Akt/Akt and p-mTOR/mTOR ratios compared to the D-gal group, then the autophagy signaling pathway was activated. However, the pFe_3_O_4_ group showed markedly decreased ratios of p-Akt/Akt and p-mTOR/mTOR compared to the control group (Figure 9).

## 3. Discussion

Long-term D-gal treatment leads to a series of pathological changes similar to natural aging, which makes this a widely accepted model for aging research [23,40]. Fe_3_O_4_ nanozymes have been shown to mimic peroxidase and catalase activities at acidic and neutral pH, respectively [32]. Previous study has shown that Fe_3_O_4_ nanozymes possess a catalase-like activity in Drosophila in vivo [33]. However, their neuroprotective effects against aging in the hippocampus have not been studied. Here, we explored the effects of PEG-Fe_3_O_4_ nanozymes on neuroblast differentiation and BBB integrity in the hippocampal DG of aging mice. The major finding of this study was that treatment with PEG-Fe_3_O_4_ nanozymes significantly promoted neuroblast differentiation in the hippocampus of D-gal-induced aged mice by increasing the levels of antioxidants and autophagy-related proteins, maintaining the integrity of BBB and inhibiting apoptosis.

### 3.1. Effects of PEG-Fe_3_O_4_ Nanozymes on Antioxidant Levels in D-Gal-Induced Aged Mice

Oxidative stress in the aging brain is a consequence of increased ROS generation and/or reduced physiological activity of antioxidant defenses (e.g., SOD, catalase, and GPX) [4]. The physiological flux of ROS regulates the cellular processes essential for cell survival, differentiation, proliferation, and migration. However, excess ROS is a hallmark of many pathological states. SOD dismutates O_2_ into H_2_O_2_ and thus prevents its accumulation to toxic levels, and then catalase converts H_2_O_2_ into H_2_O and O_2_ [41]. In the present study, PEG-Fe_3_O_4_ nanozymes were shown to increase the levels of antioxidants including SOD and catalase after treatment with D-gal. These results suggest that PEG-Fe_3_O_4_ nanozymes have neuroprotective effects via SOD and catalase-like activities. However, excess antioxidant intake will promote excessive ROS production to a level exceeding the ROS scavenging capability, resulting in a net H_2_O_2_ spillover from mitochondria, causing oxidative damage, which is called antioxidant stress [42]. Therefore, under physiological conditions, long-term pFe_3_O_4_ nanozyme intake in drinking water induces oxidative damage and leads to a series of pathological changes. Our results showed that a high dose of Fe_3_O_4_ nanozymes can still induce oxidative stress that overwhelms the ROS scavenging capability.

### 3.2. Impacts of PEG-Fe_3_O_4_ Nanozymes on BBB Integrity in D-Gal-Induced Aged Mice

The BBB is a highly complex and dynamic barrier composed of tight junctions (TJs), pericytes, astrocyte end-feet, and basal lamina [43], which is essential to protect the delicate neural tissue by regulating the homeostasis of the CNS. Dysfunction of the BBB is involved in the pathogenesis of several neurodegenerative diseases including Alzheimer’s disease and Parkinson’s disease [44,45]. Previous studies have shown that oxidative stress induced by D-gal can lead to changes in TJ proteins and BBB function [46]. Oxidative stress not only regulates the magnitude of leukocyte extravasation into the brain parenchyma but also directly causes TJ loosening, edema formation, and leakiness of the BBB in vitro and in vivo. Loss of TJ-related proteins such as Claudin5 and ZO-1 increases barrier permeability and the apoptosis of vascular endothelial cells [47,48]. Dysfunction of the endothelial cells and loss of TJs are the two main characteristics of aging microvessels. In the present study, long-term treatment with D-gal reduced the expression of PECAM-1 and BBB-related TJ proteins including the major constituents of TJ strands and cytoplasmic scaffolding (Claudin5 and ZO-1, respectively). However, the PEG-Fe_3_O_4_ nanozymes significantly suppressed D-gal-induced BBB damage, which may have been due to its antioxidant effects.

### 3.3. Effects of PEG-Fe_3_O_4_ Nanozymes on the Regulation of Autophagy in D-Gal-Induced Aged Mice

Autophagy is a critical process for the maintenance of cellular homeostasis under both normal physiological and stress conditions. Basal autophagy has emerged as a core protective mechanism against CNS aging and neurodegenerative diseases [49,50,51]. Defects in autophagy impair the proliferation and differentiation of stem cells, and decrease neurogenesis [51,52]. Beclin-1 and LC3II/I are commonly used as autophagy-related markers [53]. LC3 is involved in cargo recruitment and autophagosome biogenesis, and LC3 lipidation is routinely used as a measure of autophagy. The LC3-II/LC3-I ratio is a marker of autophagy induction in different tissues [21,50]. The E1-like enzyme, Atg7, is an essential catalyst for the transformation of LC3-I to LC3-II. Oxidative stress in aging mice prevents LC3 lipidation and impairs autophagy due to increased Atg7 oxidation [54]. Beclin-1 has a well-established role in regulating autophagy by promoting the formation of autophagic vesicles and localized autophagic proteins [55]. It has been demonstrated that LC3, Atg7, and Beclin-1 play important roles in protecting against neurodegeneration [54,56,57]. In the present study, we examined the expression of Beclin-1, LC3II/I, and Atg7 protein to evaluate the autophagy level and found that PEG-Fe_3_O_4_ nanozymes ameliorated D-gal-induced aging in the brain with deficient autophagy by upregulating the Beclin-1, LC3II/I, and Atg7 protein expression level.

The kinase, mTOR, is a major negative modulator of autophagy and the Akt pathway is an upstream major modulator of mTOR [58]. The Akt/mTOR pathway plays a pivotal role in stem cell aging [59]. Increased phosphorylation of Akt at S473 and mTOR phosphorylation at S2448 have been observed in the temporal cortex neurons in Alzheimer’s disease [60]. ROS-induced oxidative damage has been shown to activate the Akt/mTOR pathway and results in a decrease in autophagic stimulation [61,62]. Moreover, the accumulation of intracellular ROS can activate Akt and cause apoptosis [63]. Antioxidants such as coenzyme Q10 protect stem cells against aging induced by D-gal through the inhibition of the Akt/mTOR pathway [64]. We hypothesized that inactivation of Akt/mTOR signaling and decreased autophagy may be the mechanism by which PEG-Fe_3_O_4_ nanozymes inhibit brain aging induced by D-gal. Our results showed that the expression of p-Akt and p-mTOR could be deceased by PEG-Fe_3_O_4_ nanozymes.

### 3.4. Effects of PEG-Fe_3_O_4_ Nanozymes on Apoptosis in D-Gal-Induced Aged Mice

Apoptosis plays a role in the removal of aging and abnormal cells [65]. Moderate apoptosis is the process of the active termination of cells and the basic maintenance of cell homeostasis. However, excessive apoptosis activated by oxidative stress is related to the inhibition of neurogenesis [66]. In the present study, Western blotting analysis detected decreased anti-apoptotic Bcl-2 protein expression along with increased caspase-3 protein expression in D-gal-induced aged mice. PEG-Fe_3_O_4_ nanozymes decreased caspase-3 expression and increased Bcl-2 expression in D-gal-induced aged mice. These results indicate that treatment with PEG-Fe_3_O_4_ nanozymes has antiapoptotic effects in the brains of D-gal-induced aged mice. Excess PEG-Fe_3_O_4_ nanozymes decreased anti-apoptotic Bcl-2 protein expression and increased caspase-3 protein expression in the pFe_3_O_4_ group.

### 3.5. Relationships among Autophagy, Apoptosis, and BBB Integrity in D-Gal-Induced Aged Mice

Apoptosis and autophagy are highly regulated biological processes, and the balance between the two counteracting functions is essential for promoting cell viability [67]. There is a great deal of evidence that the activation of autophagy reduces or inhibits apoptosis, and vice versa [68]. The genetic inhibition of autophagy such as the disruption of Atg7 often leads to apoptosis or necrotic cell death, clearly indicating that the inhibition of autophagy leads to the activation or upregulation of apoptosis [69]. In contrast, the artificial activation of autophagy could have neuroprotective effects in the aging brain by maintaining the selective DNA damage repair pathway and the removal of ROS to inhibit apoptosis [70]. Both of these are also important for repairing damage to the BBB. There is evidence that moderate autophagy can protect endothelial cells against injury under stressful conditions and significantly reverse the decrease in level of the TJ protein, ZO-1, induced by ROS [71,72]. However, the apoptosis induced by oxidative stress was suggested to disrupt the integrity of the BBB in the aging hippocampus induced by D-galactose [46].

### 3.6. Relationships among Antioxidant, Autophagy, and Neurogenesis in D-Gal-Induced Aged Mice

Several exogenous antioxidants have been shown to improve cognitive function in aging model mice by promoting neurogenesis in the hippocampus [73,74]. Results in animals overexpressing SOD/catalase suggest that it may be possible to manipulate neurogenesis by modulating the redox state [75]. D-gal can disrupt the proliferation and differentiation of adult mouse hippocampal neuroblasts by inducing oxidative stress [76]. Ascorbic acid ameliorates D-gal induced injury and increases hippocampal neurogenesis through anti-oxidative effects [77]. Autophagy plays a key role in adult neurogenesis [78]. Autophagy promotes the proliferation and differentiation of neural stem cells by regulating reactive oxygen species and reducing intracellular oxidative stress [79]. Basal autophagy is primarily required for adequate neuronal differentiation during development [80]. Studies have shown that nanoscale electrical stimulation promotes the neurogenesis of neural stem cells by enhancing autophagy signaling [81]. Our previous research also showed that the promoted cell proliferation, neuroblast differentiation, and neural regeneration after ischemic stroke were concerned with maintaining the autophagy levels [82]. The results of this study showed that the phosphorylated expression of Akt and mTOR could be inhibited by Fe_3_O_4_ nanozymes, suggesting that Fe_3_O_4_ nanozymes might activate autophagy by inhibiting the Akt/mTOR signaling pathway to play a neuroprotective role.

In conclusion, PEG-Fe_3_O_4_ nanozymes may be effective for protection against brain aging by improving neuroblast differentiation in the hippocampal dentate gyrus. The protective mechanism may be attributable to their antioxidant-like activities, which result in the clearing of ROS, the upregulation of autophagy, the maintenance of BBB integrity, and the inhibition of apoptosis in the hippocampal DG of D-gal-induced aged mice. However, under physiological conditions, PEG-Fe_3_O_4_ nanozymes act as pro-oxidants by increasing oxidative stress. The pro-oxidant and antioxidant effects of PEG-Fe_3_O_4_ nanozymes may be dose-dependent in the aging brain, and further studies are required to determine the optimum concentration.

## 4. Materials and Methods

### 4.1. Experimental Animals

Eight-week-old male ICR mice were purchased from the Comparative Medicine Center of Yangzhou University (Yangzhou, China) and used after 1 week of acclimation. They were maintained under controlled environmental conditions with an ambient temperature of 23 °C and humidity of 60% with a 12 h light/12 h dark cycle, and were provided with free access to food and water. All experimental procedures were performed in accordance with the National Institutes of Health Guidelines for the Care and Use of Laboratory Animals. All efforts were made to minimize the number of animals used and their suffering during the experiments. The animal protocol was approved based on the ethical procedures and scientific care by the Yangzhou University-Institutional Animal Care and Use Committee (Grant No. YIACUC-15-0016).

### 4.2. Preparation and Characterization of Polyethylene Glycol (PEG)-Fe_3_O_4_ Nanozymes

With reference to a previously published procedure, the preparation and characterization of Fe_3_O_4_ nanozymes were performed [83]. PEG-Fe_3_O_4_ nanozymes were obtained by the solvothermal method and used for the synthesis of iron oxide nanoparticles. Briefly, 0.82 g of ferric chloride was dissolved in 40 mL of ethylene glycol and stirred until the solution was clear. Then, 3.6 g of sodium acetate was added to the mixture followed by 0.1 g of PEG (Mw = 10,000 Da). After stirring for 30 min to ensure the complete dissolution of all reagents, it was then sonicated for 10 min, transferred to a Teflon reactor, and sealed in an autoclave for a solvothermal reaction. After 12 h at 200 °C, the product was collected with a magnet and washed three times with ethanol to remove the unreacted reagents. The final product was aged at 60 °C for 3 h and stored in dry conditions for subsequent characterization and testing. Transmission electron microscopy (TEM; Philips CM100, 80 kV) and scanning electron microscopy (SEM; Philips XL-30 array, 15 kV) were used to describe the size distribution and morphology. The polyethylene glycol modifications were characterized by Fourier transform infrared (FTIR) spectroscopy (IRAffinity-1 spectrometer).

### 4.3. Treatment with PEG-Fe_3_O_4_ Nanozymes

The animals were divided randomly into five groups (*n* = 14 in each group): (1) the control group, 0.9% saline treatment; (2) the D-gal group, 100 mg/kg D-gal treatment; (3) the Melatonin (Mel)+D-gal group, 50 mg/kg melatonin+100 mg/kg D-gal treatment; (4) the PEG-Fe_3_O_4_ (pFe_3_O_4_) group, 10 μg/mL PEG-Fe_3_O_4_ nanozymes treatment; and (5) the pFe_3_O_4_+D-gal group, 10 μg/mL PEG-Fe_3_O_4_ nanozymes+100 mg/kg D-gal treatment. Melatonin is a positive endogenous regulator of neurogenesis, and plays important roles in memory improvement and promoting neurogenesis in the mouse hippocampus during normal aging [84]. In the present study, melatonin was used as a positive control. D-gal was administered intraperitoneally at a dose of 100 mg/kg once per day for 12 weeks. Based on previous studies, melatonin was administered at a dose of 50 mg/kg body weight by gavage once daily for the last 4 weeks. Fe_3_O_4_ nanozymes (10 μg/mL diluted in ddH_2_O) were added to the drinking water for the last 4 weeks. We also used PEG, a polymer that can improve the biological safety, prolong the biological half-life in vivo, and increase the permeability of nanoparticles in the brain by prolonging the blood circulation time to modify iron oxide nanoparticles in a solvothermal synthetic process. The animals were weighed twice a week during the drug treatment.

### 4.4. Tissue Processing for Histology

For histological analysis, the animals were narcotized with 1.25% 2,2,2-tribromoethanol (Sigma, St. Louis, MO, USA) and perfused transcardially with 0.1 M phosphate-buffered saline (PBS, pH 7.4) followed by 4% paraformaldehyde in 0.1 M phosphate buffer (PB, pH 7.4). The brains were removed and postfixed in 4% paraformaldehyde for 24 h before cryoprotection by infiltration with 30% sucrose overnight. Thereafter, frozen tissues were cut coronally into sections at a thickness of 30 μm on a cryostat (Leica, Wetzlar, Germany) and stored in six-well plates containing PBS.

### 4.5. Measurement of SOD Activity and MDA Content

With reference to the previously published procedure, the SOD activity and MDA content were measured [83]. The brain tissue of the experimental mouse model was taken out, placed in a 4 °C refrigerator overnight, centrifuged at 3000 rpm for 10 min, and the supernatant was collected. The SOD activity and MDA content of mice in each group were detected by the xanthine oxidase method and thiobarbituric acid method, respectively. The SOD activity was detected using the WST-1 cell proliferation assay kit, while the MDA content in the homogenate was measured using the total bile acid colorimetric method according to the manufacturer’s instructions.

### 4.6. Immunohistochemistry

The sections were treated sequentially with 0.3% hydrogen peroxide (H_2_O_2_) in PBS for 20 min and 5% normal serum in 0.01 M PBS for 30 min. Next, they were incubated with diluted goat anti-doublecortin (DCX) antibody (1:200; Santa Cruz, Santa Cruz, CA, USA), rabbit anti-Ki-67 antibody (1:250; Abcam, Cambridge, MA, USA), and mouse anti-platelet endothelial cell adhesion molecule-1 (PECAM-1) antibody (1:500; Bio-Techne, Minneapolis, MN, USA) overnight at 4 °C. Thereafter, they were exposed to biotinylated rabbit anti-goat, goat anti-rabbit, or goat anti-mouse IgG as appropriate (1:250; Vector Laboratories, Burlingame, CA, USA) and streptavidin peroxidase complex (1:200; Vector Laboratories). In addition, they were visualized with 3,3′-diaminobenzidine tetrahydrochloride in 0.01 M PBS and mounted on Adhesion Microscope slides. After dehydration, the sections were sealed with neutral balsam (Solarbio, Beijing, China). To establish the specificity of immunological staining, pre-immune serum was used instead of the primary antibody as a negative control and showed no immunoreactivity in any structures. Digital images of the hippocampal subregions were captured using an image analysis system equipped with a computer-based microscope (Nikon Corporation, Tokyo, Japan). The structures showing DCX and PECAM-1 immunoreactivity in the DG were examined at ×200 magnification, and the figure of the DCX immunohistochemistry was examined at ×400 magnification. Cell counts were determined by averaging the counts of sections from each animal. Furthermore, the staining intensity of the PECAM-1 immunoreactive structures was evaluated based on optical density (OD) after average gradation conversion using the following formula: OD  =  log (256/average grayscale). The OD of the background was taken from areas adjacent to the measured area. After subtracting the background density, the OD ratio was calibrated to % (relative OD, ROD) using Photoshop version 8.0 (Adobe Systems Inc., San Jose, CA, USA), and then analyzed using NIH Image 1.59 software (NIH, Bethesda, MD, USA). All measurements were taken under blinded conditions to ensure objectivity, and each experiment was performed by two observers under the same conditions.

### 4.7. Western Blotting Analysis

Experimental animals (*n*  =  7 per group) were used for Western blotting analysis. After sacrifice, the brain was removed and cut serially and transversely into slices 400 μm thick on a vibrating microtome, and the hippocampus was dissected with a surgical blade. The tissues were pretreated using a Whole Cell Lysis Assay Kit/Total Protein Extraction Kit (KeyGEN, Nanjing, China). Protein concentrations were determined using a BCA Protein Assay Kit (Thermo Fisher Scientific, Rockford, IL, USA). Briefly, aliquots of 30 µg protein were separated by 10% sodium dodecyl sulfate-polyacrylamide gel electrophoresis (SDS-PAGE) and electrotransferred onto polyvinyl difluoride (PVDF) membranes (Millipore, Bedford, MA, USA). Membranes were blocked with 5% (*w*/*v*) skim milk or 5% (*w*/*v*) bovine serum albumin (BSA) in Tris-buffered saline (TBS) containing 0.1% Tween 20 (TBST) for 2 h at room temperature to reduce the background staining. Then, the membranes were incubated with rabbit antibodies to zonula occludens-1 (ZO-1) (1:1000; Thermo Fisher Scientific), Claudin5 (1:1000; Abcam, Cambridge, MA, USA), SOD1 and SOD2 (1:1000; Arigo, Burlington, ON, Canada), catalase (1:1000; Abcam), Bcl-2 (1:1000; Santa Cruz), caspase-3 (1:1000; Cell Signaling Technology, Danvers, MA, USA), cleaved caspase-3 (1:1000; Cell Signaling Technology), Beclin-1 (1:1000; Santa Cruz), microtubule-associated Protein 1A/1B Light chain 3 (LC3II/I) (1:1000; Cell Signaling Technology), Autophagy Related 7 (Atg7) (1:1000; Cell Signaling Technology), protein kinase B (Akt) (1:1000; Cell Signaling Technology), phospho (p)-Akt (1:1000; Cell Signaling Technology), mammalian target of rapamycin (mTOR) (1:1000; Cell Signaling Technology), p-mTOR (1:1000; Santa Cruz), and β-actin (1:1000; Arigo) overnight at 4 °C with gentle agitation and washed three times for 5 min each time with 15 mL TBST, and then subsequently exposed to secondary goat anti-rabbit IgG (Santa Cruz) at room temperature for 2 h, and Super Signal West Pico Chemiluminescent Substrate (Thermo Fisher Scientific) was used for protein detection. The results of the Western blotting analysis were scanned and subjected to quantitative OD analysis using Quantity One Analysis Software (Bio-Rad, Hercules, CA, USA) to calculate the ratio of ROD: calibration ROD in %, with the control group designated as 100%. Each blot shown is representative of at least three similar independent experiments.

### 4.8. Fe_3_O_4_ Nanozyme Concentrations in the Mouse Brain

To quantify the Fe_3_O_4_ nanozymes in the normal mouse brain after administration in the drinking water (10 μg/mL PEG-Fe_3_O_4_ nanozymes) for 0 h, 24 h, 48 h, and 72 h, we measured the content of iron in the brain. The brain tissues were lyophilized and mashed prior to acid digestion. Samples were incubated with 2 mL aqua regia (a 1:3 mixture of nitric acid and hydrochloric acid) for 24 h at 90 °C. The iron content was analyzed by inductively coupled plasma optical emission spectroscopy (ICP-OES).

### 4.9. Statistical Analysis

All experimental data were statistically processed using SPSS 27.0 software, and all data were expressed as the mean ± SEM. Differences in the means among the groups were analyzed by one-way analysis of variance (ANOVA) with Duncan’s post hoc test. In all analyses, *p* < 0.05 was taken to indicate statistical significance.

## Figures and Tables

**Figure 1 ijms-23-06463-f001:**
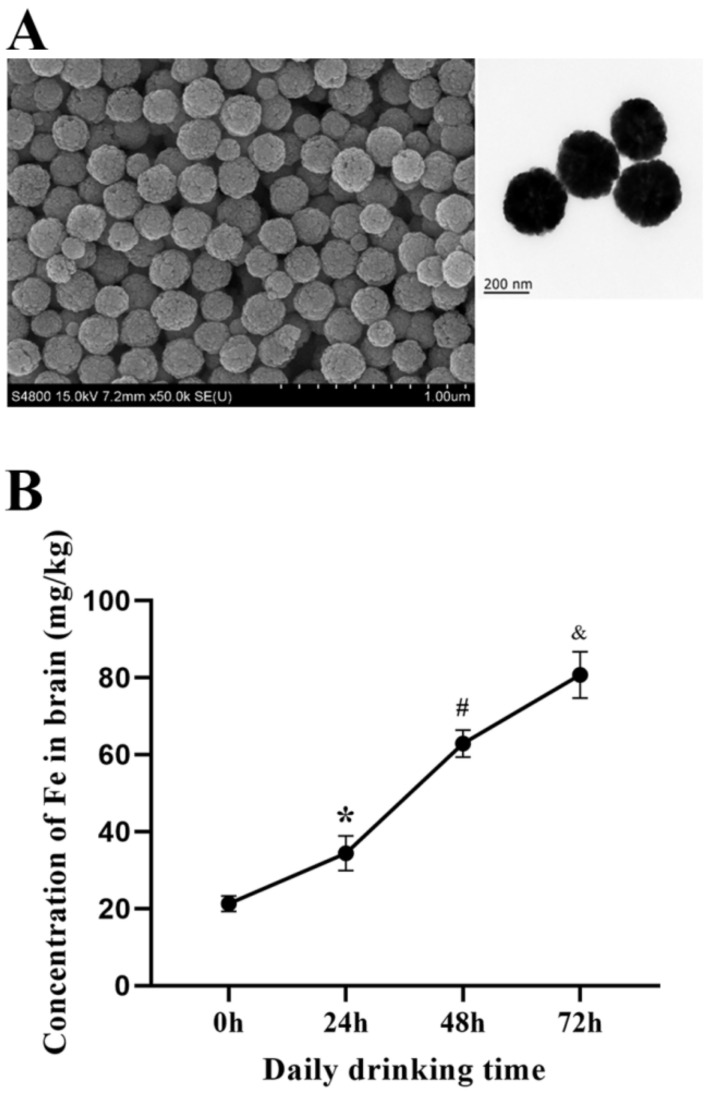
(**A**) PEG-Fe_3_O_4_ nanozymes under scanning electron microscope and transmission electron microscope. (**B**) PEG-Fe_3_O_4_ nanozyme concentrations in the brains of mice at different time points. Compared to 0 h, the concentration of iron in the brain tissue of mice increased after 24 h (* *p* < 0.05), increased significantly after 48 h compared to 24 h (^#^
*p* < 0.05), and further increased significantly after 72 h compared to 48 h (^&^
*p* < 0.05).

**Figure 2 ijms-23-06463-f002:**
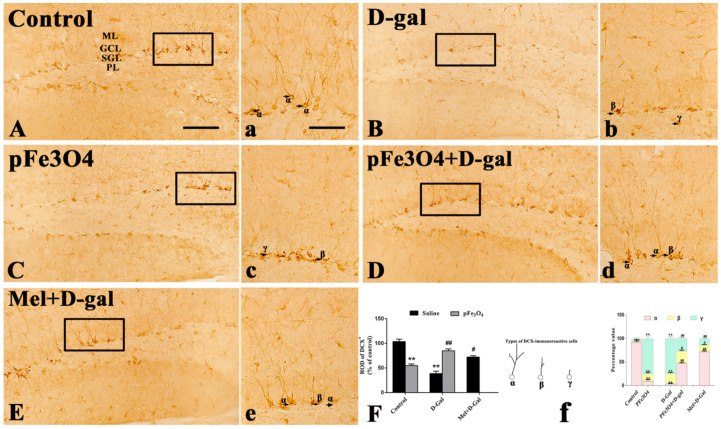
Immunohistochemical analysis of doublecortin (DCX) in the hippocampal DG of: (**A**) the control group (0.9% saline treatment); (**B**) the D-gal group (100 mg/kg D-gal treatment); (**C**) the pFe_3_O_4_ group (10 μg/mL PEG-Fe_3_O_4_ nanozymes treatment); (**D**) the pFe_3_O_4_+D-gal group (10 μg/mL PEG-Fe_3_O_4_ nanozymes+100 mg/kg D-gal treatment); and (**E**) the Mel+D-gal group (50 mg/kg melatonin+100 mg/kg D-gal treatment). PL, polymorphic layer; GCL, granule cell layer; SGZ, sub-granular zone; ML, molecular layer. Scale bar: 50 μm. (**F**): The number of DCX-immunoreactive cells per section in different groups (*n* = 7 per group; ** *p* < 0.01, vs. the control group; ^#^
*p* < 0.05, ^##^
*p* < 0.01 vs. the D-gal group; ANOVA with Duncan’s post hoc test).DCX^+^ processes (arrows) in the DG of: (**a**) the control group; (**b**) the D-gal group; (**c**) the pFe_3_O_4_ group; (**d**) the pFe_3_O_4_+D-gal group; (**e**) the Mel+D-gal group. (**f**) Analysis of dendrite category to examine changes in dendritic complexity of DCX-immunoreactive cells and the percentage values of types “α”, “β”, and “γ” of DCX-immunoreactive cells per section in the DG (“α” indicates cells with many more branches reaching the ML, “β” indicates cells with one primary dendrite with one branch, “γ” indicates cells without dendrites or with dendrites shorter than the soma size). Scale bar: 25 μm. Data are presented as the means ± SEMs (*n* = 7 per group; ** *p* < 0.01 vs. the control group; ^#^
*p* < 0.05, ^##^
*p* < 0.01 vs. the D-gal group; ANOVA followed by Tukey’s multiple range test.

**Figure 3 ijms-23-06463-f003:**
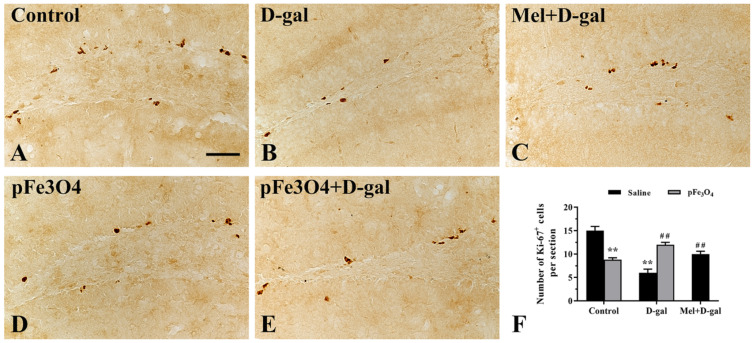
Immunohistochemical analysis of Ki-67 in the hippocampal DG of: (**A**) the control group (0.9% saline treatment); (**B**) the D-gal group (100 mg/kg D-gal treatment); (**C**) the Mel+D-gal group (50 mg/kg melatonin+100 mg/kg D-gal treatment); (**D**) the pFe_3_O_4_ group (10 μg/mL PEG-Fe_3_O_4_ nanozymes treatment); and (**E**) the pFe_3_O_4_+D-gal group (10 μg/mL PEG-Fe_3_O_4_ nanozymes+100 mg/kg D-gal treatment). Scale bar: 50 μm. (**F**): The number of Ki-67-immunoreactive cells per section in different groups (*n* = 7 per group; ** *p* < 0.01, vs. the control group; ^##^
*p* < 0.01 vs. the D-gal group; ANOVA with Duncan’s post hoc test).

**Figure 4 ijms-23-06463-f004:**
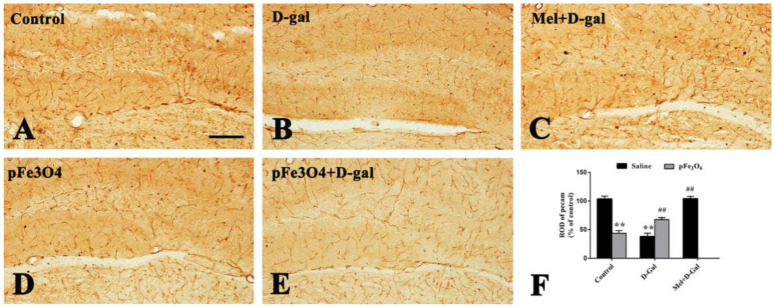
PECAM-1 immunoreactivity cells in the hippocampal DG of: (**A**) the control group (0.9% saline treatment); (**B**) the D-gal group (100 mg/kg D-gal treatment); (**C**) the Mel+D-gal group (50 mg/kg melatonin+100 mg/kg D-gal treatment); (**D**) the pFe_3_O_4_ group (10 μg/mL PEG-Fe_3_O_4_ nanozymes treatment); and (**E**) the pFe_3_O_4_+D-gal group (10 μg/mL PEG-Fe_3_O_4_ nanozymes+100 mg/kg D-gal treatment). Scale bars: 50 μm. Data are presented as the means ± SEMs. (**F**) The number of PECAM-1 immunoreactivity cells per section in different groups (*n* = 7 per group; ** *p* < 0.01 vs. the control group; ^##^
*p* < 0.01 vs. the D-gal group; ANOVA with Duncan’s post hoc test).

**Figure 5 ijms-23-06463-f005:**
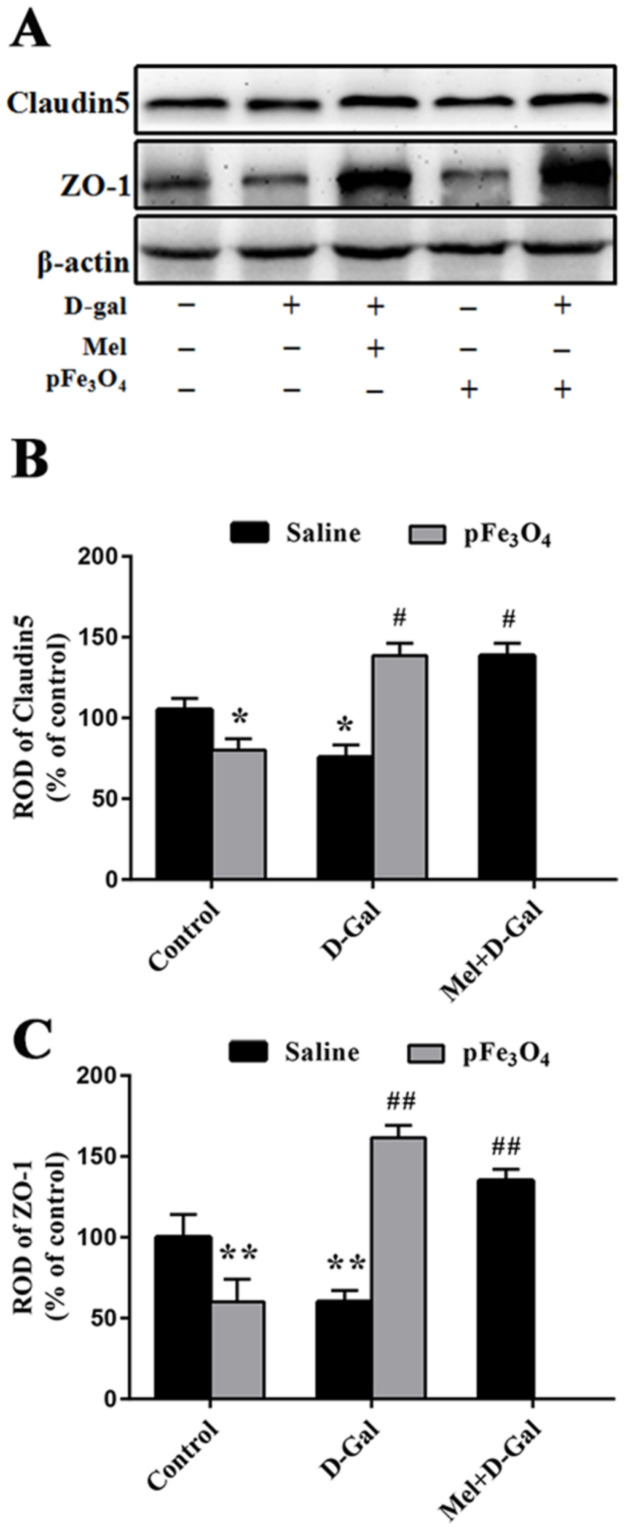
(**A**) Western blotting analysis of Claudin5 and ZO-1 in the hippocampus of the control group (D-gal-, Mel-, pFe_3_O_4_-); 100 mg/kg D-gal-treated group (D-gal+, Mel-, pFe_3_O_4_-); 50 mg/kg Melatonin+100 mg/kg D-gal-treated group (D-gal+, Mel+, pFe_3_O_4_-); 10 μg/mL PEG-Fe_3_O_4_ nanozyme-treated group (D-gal-, Mel-, pFe_3_O_4_+); 10 μg/mL PEG-Fe_3_O_4_ nanozymes+100 mg/kg D-gal-treated group (D-gal+, Mel-, pFe_3_O_4_+). (**B**,**C**) ROD presented as percentages of the immunoblot band (*n* = 7 per group; * *p* < 0.05, ** *p* < 0.01 vs. the control group; ^#^
*p* < 0.05, ^##^
*p* < 0.01 vs. the D-gal group). Bars indicate the means ± SDs.

**Figure 6 ijms-23-06463-f006:**
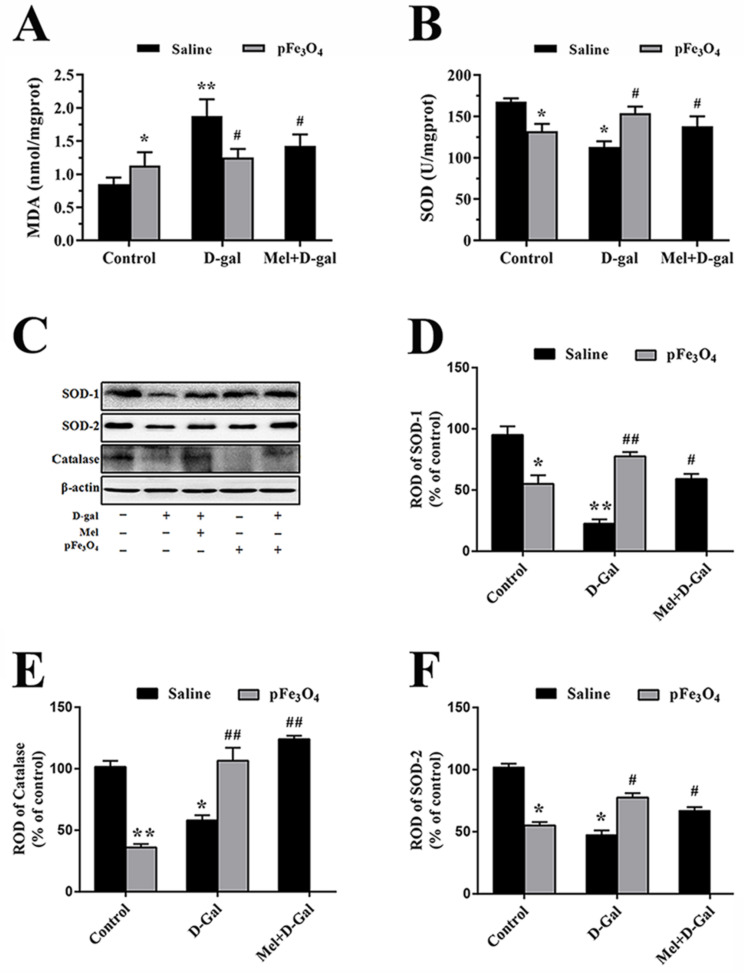
(**A**) Measurement of the MDA content. (**B**) Measurement of SOD activity. (**C**) Western blotting analysis of SOD1, SOD2, and catalase in the hippocampus of the control group (D-gal-, Mel-, pFe_3_O_4_-); 100 mg/kg D-gal-treated group (D-gal+, Mel-, pFe_3_O_4_-); 50 mg/kg Melatonin+100 mg/kg D-gal-treated group (D-gal+, Mel+, pFe_3_O_4_-); 10 μg/mL PEG-Fe_3_O_4_ nanozyme-treated group (D-gal-, Mel-, pFe_3_O_4_+); 10 μg/mL PEG-Fe_3_O_4_ nanozymes+100 mg/kg D-gal-treated group (D-gal+, Mel-, pFe_3_O_4_+). (**D**–**F**) ROD presented as percentages of the immunoblot band (*n* = 7 per group; * *p* < 0.05, ** *p* < 0.01 vs. the control group; ^#^
*p* < 0.05, ^##^
*p* < 0.01 vs. the D-gal group). Bars indicate the means ± SDs.

**Figure 7 ijms-23-06463-f007:**
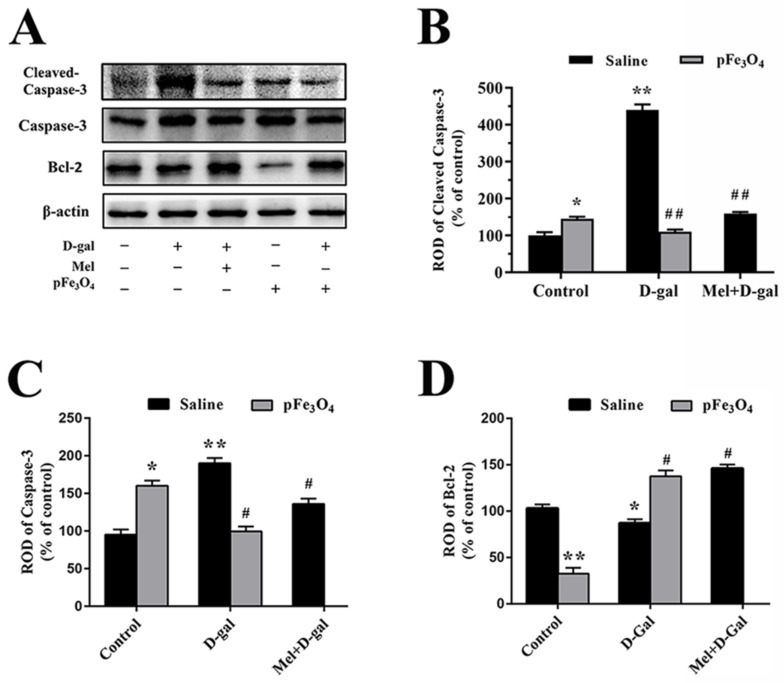
(**A**) Western blotting analysis of cleaved caspase-3, caspase-3, and Bcl-2 in the hippocampus of the control group (D-gal-, Mel-, pFe_3_O_4_-); 100 mg/kg D-gal-treated group (D-gal+, Mel-, pFe_3_O_4_-); 50 mg/kg Melatonin+100 mg/kg D-gal-treated group (D-gal+, Mel+, pFe_3_O_4_-); 10 μg/mL PEG-Fe_3_O_4_ nanozyme-treated group (D-gal-, Mel-, pFe_3_O_4_+); 10 μg/mL PEG-Fe_3_O_4_ nanozymes+100 mg/kg D-gal-treated group (D-gal+, Mel-, pFe_3_O_4_+). (**B**–**D**) ROD presented as percentages of the immunoblot band (*n* = 7 per group; * *p* < 0.05, ** *p* < 0.01 vs. the control group; ^#^
*p* < 0.05, ^##^
*p* < 0.01 vs. the D-gal group). Bars indicate the means ± SDs.

**Figure 8 ijms-23-06463-f008:**
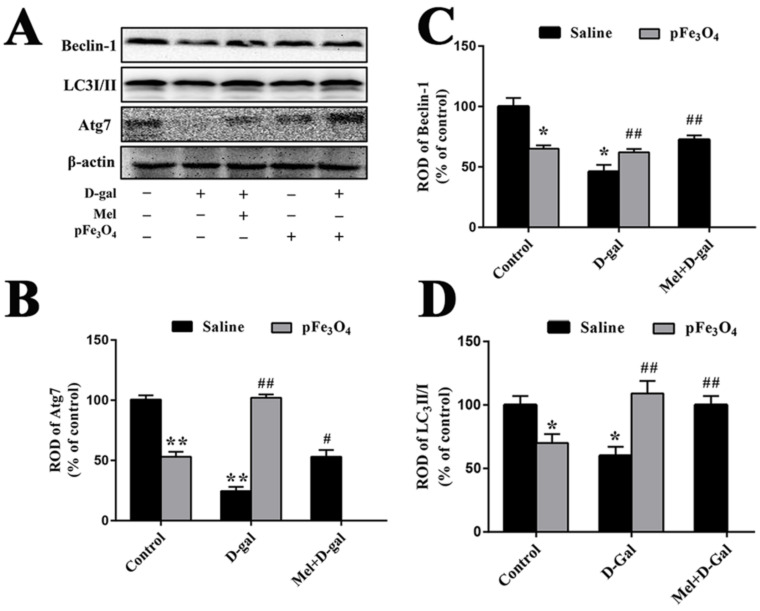
(**A**) Western blotting analysis of Beclin-1, LC3II/I, and Atg7 in the hippocampus of the control group (D-gal-, Mel-, pFe_3_O_4_-); 100 mg/kg D-gal-treated group (D-gal+, Mel-, pFe_3_O_4_-); 50 mg/kg Melatonin+100 mg/kg D-gal-treated group (D-gal+, Mel+, pFe_3_O_4_-); 10 μg/mL PEG-Fe_3_O_4_ nanozyme-treated group (D-gal-, Mel-, pFe_3_O_4_+); 10 μg/mL PEG-Fe_3_O_4_ nanozymes+100 mg/kg D-gal-treated group (D-gal+, Mel-, pFe_3_O_4_+). (**B**–**D**) ROD presented as percentages of the immunoblot band (*n* = 7 per group; * *p* < 0.05, ** *p* < 0.01 vs. the control group; ^#^
*p* < 0.05, ^##^
*p* < 0.01 vs. the D-gal group). Bars indicate the means ± SDs.

**Figure 9 ijms-23-06463-f009:**
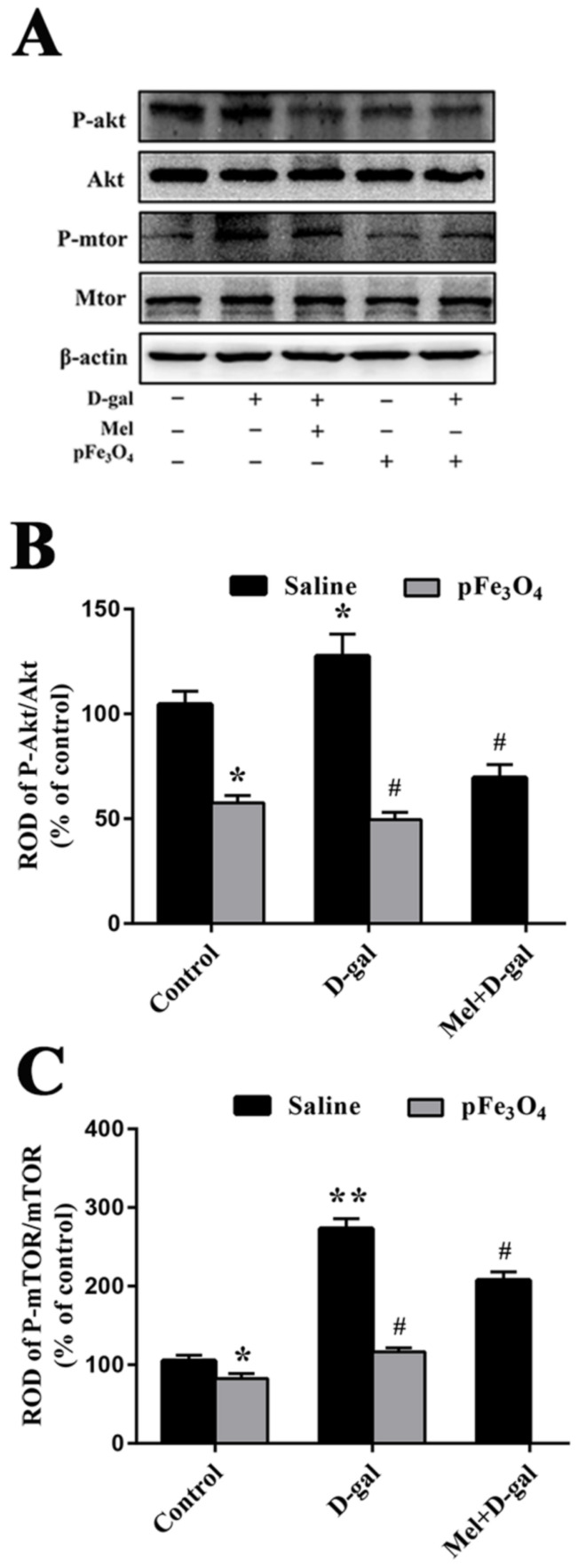
(**A**) Western blotting analysis of the ratios of p-Akt/Akt and p-mTOR/mTOR in the hippocampus of the control group (D-gal-, Mel-, pFe_3_O_4_-); 100 mg/kg D-gal-treated group (D-gal+, Mel-, pFe_3_O_4_-); 50 mg/kg Melatonin+100 mg/kg D-gal-treated group (D-gal+, Mel+, pFe_3_O_4_-); 10 μg/mL PEG-Fe_3_O_4_ nanozyme-treated group (D-gal-, Mel-, pFe_3_O_4_+); 10 μg/mL PEG-Fe_3_O_4_ nanozymes+100 mg/kg D-gal-treated group (D-gal+, Mel-, pFe_3_O_4_+). (**B**,**C**) ROD presented as percentages of the immunoblot band (*n* = 7 per group; * *p* < 0.05, ** *p* < 0.01 vs. the control group; ^#^
*p* < 0.05 vs. the D-gal group). Bars indicate the means ± SDs.

## Data Availability

All data generated or analyzed during this study are available from the corresponding author on reasonable request.

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
