# Peer review of "Fe3O4 Nanozymes Improve Neuroblast Differentiation and Blood-Brain Barrier Integrity of the Hippocampal Dentate Gyrus in D-Galactose-Induced Aged Mice"

_ijms, 2022, doi:10.3390/ijms23126463_

Round 1
Reviewer 1 Report
The mauscript by Xia et al., was intended to examined the effects of Fe3O4 nanozymes on neuronal differentiation in the dentate gyrus (DG) and BBB integrity of D-galactose-induced aged mice. The Authors conclude that Fe3O4 nanozymes can relieve neuroblast damage and promote neuroblast differentiation in the hippocampal DG by regulating oxidative stress, apoptosis and autophagy.
There are a series of poit to be addressed before the manuscript is suitable for publication.
Major points:
- To detect neurogenesis it is no more sufficient to stain for DCX, since it has been shown that DCX+ neurons that are not newly generated can be found in the adult brain (La Rosa et al., 2020, Front Neurosci; Piumatti et al., 2018, J Neurosci). These cells cam be generated prenatally and then retain DCX expression for years maintaining a state of immaturity. Otherwise the DCX+ cells can slow down their maturation being not generated recently. Although most of these cells were found in other regions (e.g. cerebral cortex), there are many suggestions that they can be also present in the neurogenic niches.
In other words, the “increase in immunoreactivity for DCX is not directly a sign for increased neurogenesis. Even aging can increase the expression of DCX (without affecting neurogenesis) in the process of “dematuration” (Hagihara et al., 2019, Mol Brain). That of DCX as a proxy for neurogenesis is an old view; most studies, in order to show neurogenesis perform at least double staining with either BrdU or Ki-67 antigen, by using immunofluorescence (and possibly confocal microscopy).
- There is also a mix and a confusion between the process of neurogenesis and the differentiation of neuroblasts (the latter should be assessed by using a set of additional markers). This confusion must be resolved.
Minor point:
- Photographs in Fig. 2 and 3 are very dark. The background is yellow-orange and the staining for DCX and PECAM-1 is not enough contrasted/emerging from the backgroung.
I think this is aproblem of taking photos from the light microscopy and, rhen, to process them with graphicaal softwares.
Author Response
I really appreciate reviewer’ prudent comments for our manuscript.
Please see the attachment.

Reviewer 2 Report
In this manuscript, Xia et al. determined protective and neuroblast differentiation promotion activity of Fe2O3 nanozymes in the subgranular zone (SGZ) of the hippocampal dentate gyrus (DG) using D-galatoce-induced aging mice model. They show that iron content in the Fe2O3 treated mice brain increased suggesting penetration of Fe2O3 nanozymes into brain. Using immunohistochemical analysis, author show that Fe2O3 increased DCX+ cells suggesting neurogenesis and increased expression of Pecam-1, Claudin5 and ZO-1 suggesting increased BBB integrity. Further they use western blot analysis to show increase in antioxidant and antiapoptotic markers with Fe2O3 tratement in D-galactose-induced aging mice. Overall, the paper is well written but the data presented in not convincing. Further, authors did not report on the discrepancy in the effect of Fe2o3 in the current manuscript compared to their recent publication determining protective effect of Fe2O3 nanozymes in ICS. Also, authors have not discussed or cited their previous extremely relevant publication in this manuscript, or discussed preparation, characterization, or source of Fe2O3 nanozymes.
Major comments:
1. For all the experiment conducted in the study, Fe2o3 on its own had same effect as D-glactose and quite opposite to control group. Further, Fe2O3 reversed the effects of D-galactose. The explanation provided by the authors ‘under physiological conditions, long-term pFe3O4 nanozyme intake in drinking water induces oxidative damage and leads to a series of pathological changes. Our results are consistent with a previous study showing that a high dose of Fe3O4 nanozymes can still induce oxidative stress that overwhelms the ROS scavenging capability [33].’ cannot be applied because the cited study to do not show that. Also, in the cited study (ref 33) even at a high dose (100ug/ml), Fe2O3 did not induce apoptosis, caspase3 and the cell viability was same as control group. Also, in recent publication by the authors ‘Dietary Fe 3 O 4 Nanozymes Prevent the Injury of Neurons and Blood-Brain Barrier Integrity from Cerebral Ischemic Stroke’ (link: https://pubmed.ncbi.nlm.nih.gov/33346645/ ) (which is not cited in the current manuscript) at a dose of Fe2O3 similar to current manuscript, there was no major difference in Fe2O3 vs control.
2. Further the results western blot results are not convincing enough: especially blots for Claudin5, Catalase (original blot missing), SOD-1, BCL2, Beclin1, and p-AKT. All quantifications show error bars but there is only 1 original blot presented in the original paper making interpretation hard. Further, there are betters assays and markers for many of these parameters which would make data more convincing and easy to interpret. Cell proliferation (Ki67), Cleaved Caspase-3 instead of Caspase 3 (apoptosis), measuring antioxidant levels instead of western blot levels of proteins, etc.
3. Most of the experiments are not novel (same experiments have been done with their previous study (not cited) as well as by others (reference 33) ). Only new thing is the model used for the study. Further observations between markers studied (antioxidant, autophagy and Akt and mTOR) and effect of Fe2o3 on neurogenesis are association. Nothing in the study shows a causation.
4. Looking at the Figure 1 and Figure3, there seems hardly any difference between Fe2O3 and Fe2O3 +D-gal group, making the effect of minor at best. Further, there is no mention in the manuscript about the preparation and characterization of Fe2O3 nanozymes.
Author Response

(The authors gave the same response as above.)

Round 2
Reviewer 1 Report
The manuscript has been appropriately revised
Reviewer 2 Report
The author have made sufficient changes to accept the manuscript. Though the conclusions are mostly correlation.
This manuscript is a resubmission of an earlier submission. The following is a list of the peer review reports and author responses from that submission.